# Homocysteine is not a risk factor for subclinical coronary atherosclerosis in asymptomatic individuals

Sangwoo Park[1], Gyung-Min Park[1]*, Jinhee Ha[2]*, Young-Rak Cho[3], Jae-Hyung Roh[4], Eun Ji Park[5], Yujin Yang[1,6], Ki-Bum Won[1], Soe Hee Ann[1], Yong-Giun Kim[1], Shin-Jae Kim[1], Sang-Gon Lee[1], Dong Hyun Yang[7], Joon-Won Kang[7], Tae-Hwan Lim[7], Hong-Kyu Kim[8], Jaewon Choe[8], Seung-Whan Lee[6], Young-Hak Kim[6]

1 Department of Cardiology, Ulsan University Hospital, University of Ulsan College of Medicine, Ulsan, Republic of Korea, 2 Department of Dentistry, Ulsan University Hospital, University of Ulsan College of Medicine, Ulsan, Republic of Korea, 3 Department of Cardiology, Dong-A University Hospital, Busan, Republic of Korea, 4 Division of Cardiology, Department of Internal Medicine, Chungnam National University Hospital, Chungnam National University School of Medicine, Daejeon, Republic of Korea, 5 Medical Information Center, Ulsan University Hospital, University of Ulsan College of Medicine, Ulsan, Republic of Korea, 6 Department of Cardiology, Asan Medical Center, University of Ulsan College of Medicine, Seoul, Republic of Korea, 7 Department of Radiology, Asan Medical Center, University of Ulsan College of Medicine, Seoul, Republic of Korea, 8 Health Screening and Promotion Center, Asan Medical Center, University of Ulsan College of Medicine, Seoul, Republic of Korea

* min8684@hanmail.net (GMP); 0732289@uuh.ulsan.kr (JH)

## Abstract

### Background

Homocysteine has been known as a risk factor for cardiovascular disease. This study sought to evaluate the influence of homocysteine on the risk of subclinical coronary atherosclerosis in asymptomatic individuals.

### Methods

We reviewed 3,186 asymptomatic individuals (mean age 53.8 ± 8.0 years, 2,202 men [69.1%]) with no prior history of coronary artery disease who voluntarily underwent coronary computed tomographic angiography (CCTA) and laboratory tests as part of a general health examination. The subjects were stratified into tertiles according to their homocysteine levels. The degree and extent of subclinical coronary atherosclerosis were assessed by CCTA. Logistic regression analysis was used to determine the association between homocysteine levels and subclinical coronary atherosclerosis.

### Results

The prevalence of significant coronary artery stenosis, any atherosclerotic, calcified, mixed, and non-calcified plaques increased with homocysteine tertiles (all p < 0.05). However, after adjustment for cardiovascular risk factors, there were no statistically significant differences in the adjusted odds ratios (ORs) for any atherosclerotic plaque (OR 1.06; 95% CI [confidence interval] 0.85–1.32; p = 0.610), calcified plaques (OR 1.17; 95% CI 0.92–1.48; p =

---

**Data Availability Statement:** Data cannot be shared publicly because data contain potentially identifying or sensitive patient information. Interested readers can request the data from the

data access committee (contact via seungwlee@amc.seoul.kr), or they can also contact the corresponding authors (min8684@hanmail.net or 0732289@uuh.ulsan. kr). As the original dataset contains variables coded in Korean, interested readers may specifically request a recorded version of the data set when requesting the data as an aid to performing analysis.

**Funding:** This research was supported by Basic Science Research Program through the National Research Foundation of Korea funded by the Ministry of Education (Grant number: 2018R1D1A3B07043344) The funders had no role in study design, data collection and analysis, decision to publish, or preparation of the manuscript.

**Competing interests:** The authors have declared that no competing interests exist.

**Abbreviations:** CACS, coronary artery calcium score; CAD, Coronary artery disease; CCTA, coronary computed tomography angiography; CI, confidence interval; CRP, C-reactive protein; CVD, cardiovascular disease; Hcy, homocysteine; OR, odds ratio; RR, risk ratio.

0.199), non-calcified plaques (OR 0.80; 95% CI 0.61–1.04; p = 0.089), and mixed plaques (OR 1.42; 95% CI 0.96–2.11; p = 0.077) between the third and first homocysteine tertiles. In addition, the adjusted OR for significant coronary artery stenosis (OR 0.92; 95% CI 0.63–1.36; p = 0.687) did not differ between the first and third tertiles.

## Conclusions

In asymptomatic individuals, homocysteine is not associated with an increased risk of subclinical coronary atherosclerosis.

## Introduction

Coronary artery disease (CAD) is the leading cause of death globally [1]. Although three quarters of CAD risk is explained by traditional cardiovascular risk factors, significant residual risk for CAD still exists in the population independent of these risk factors [2,3]. Early observational studies and meta-analysis indicated that homocysteine (Hcy) is a novel risk factor for cardiovascular disease (CVD) [4–6]. However, following studies showed conflicting results on reduction of CVD events with Hcy-lowering therapy of B vitamins [7–10]. Several randomized trials did not demonstrate the clinical benefit of Hcy-lowering therapy in CAD prevention [8,11–13]. In addition, a recent meta-analysis has not shown an association of Hcy-lowering treatment with the incidence of all-cause death or CAD, except a modest prevention of stroke [14]. Therefore, it remains unclear whether Hcy is a risk factor for CAD. Furthermore, there are limited data regarding whether the Hcy is associated with subclinical coronary atherosclerosis in asymptomatic individuals. With the advent of multidetector row computed tomography, coronary computed tomography angiography (CCTA) has proven to be effective in evaluating coronary atherosclerosis and enabled to identify and determine the relationship between cardiovascular risk factors and coronary atherosclerosis [15,16]. Therefore, this study sought to evaluate the impact of Hcy on the risk of subclinical coronary atherosclerosis in a large cohort of asymptomatic Korean individuals who voluntarily underwent CCTA for early detection of CAD.

## Methods

### Study population

A total of 9,269 consecutive South Korean individuals aged 20 years and older had undergone self-referral CCTA evaluation as part of a general health examination in the Health Screening and Promotion Center at Asan Medical Center between January 2007 and December 2011. Among them, 7,129 (76.9%) subjects agreed to participate in the current study. Possible risks associated with CCTA were explained and written informed consent was obtained. We excluded subjects with 1) unmeasured Hcy; 2) a previous history of angina or myocardial infarction; 3) abnormal rest electrocardiographic results, i.e., pathological Q waves, ischemic ST segments or T wave changes, or left bundle-branch blocks; 4) incomplete medical records; 5) structural heart diseases, i.e., hypertrophic cardiomyopathy, moderate to severe valvular heart disease, atrial septal defect, dilated cardiomyopathy, myxoma, or dextrocardia; 6) a prior history of any open heart surgery or percutaneous coronary intervention; 7) a previous cardiac procedure; or 8) renal insufficiency (creatinine > 1.5 mg/dL). Finally, 3,186 subjects were enrolled (Fig 1). This study was approved by the local Institutional Review Board of the Asan

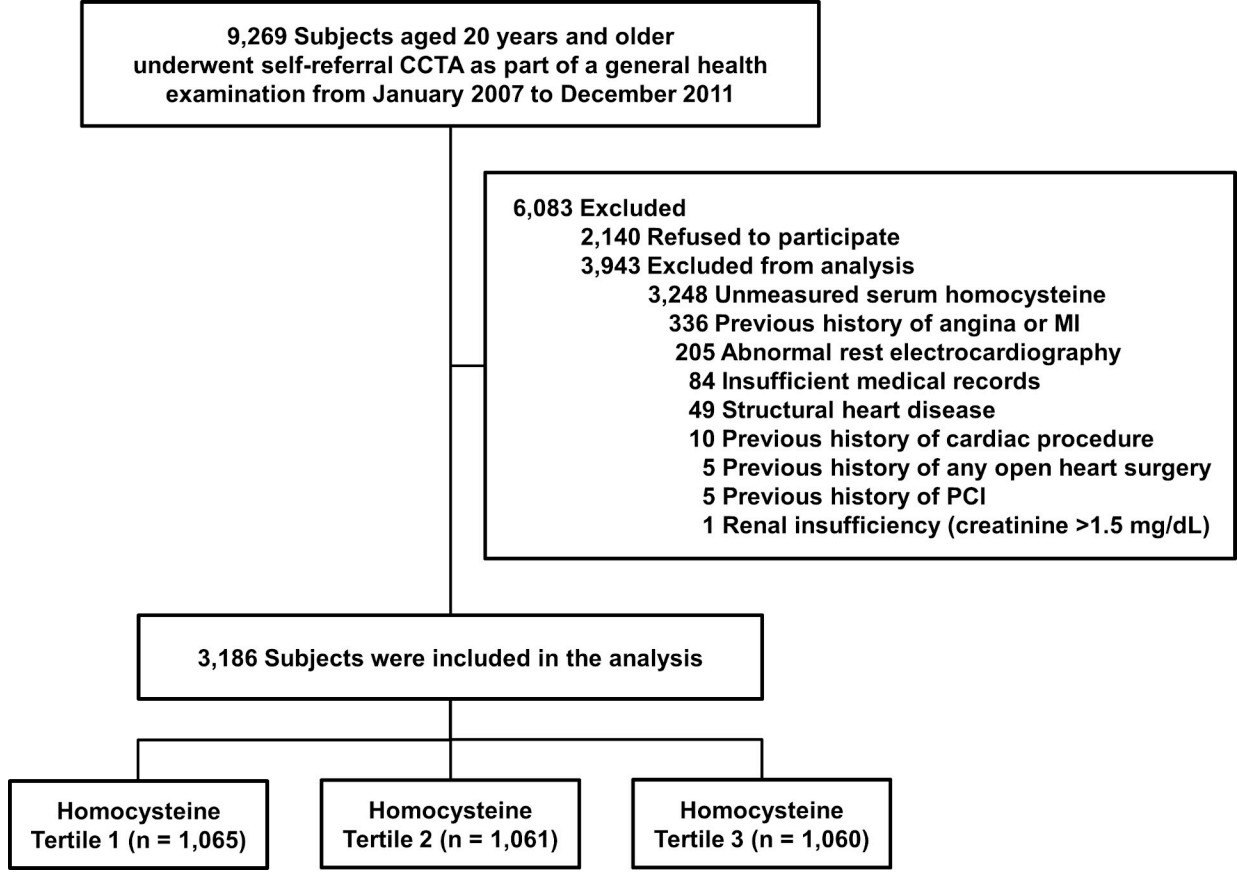

**Fig 1. Overview of the study population.** CCTA = coronary computed tomographic angiography; MI = myocardial infarction; PCI = percutaneous coronary intervention.

Medical Center, Seoul, Republic of Korea. This study conformed to the ethical guidelines outlined in the Declaration of Helsinki.

## Clinical and laboratory measurements

The basic demographic information was collected from a database maintained by the Health Screening and Promotion Center at the Asan Medical Center. Past medical history including angina, myocardial infarction, stroke, structural heart disease, open heart surgery, percutaneous coronary intervention, previous cardiac procedures, diabetes mellitus, hypertension, or hyperlipidemia; a family history of CAD; and smoking status was taken from the responses in the systemized self-report questionnaire issued prior to the general health examination [17].

Height and weight were obtained while subjects wore light clothing without shoes. The body mass index was calculated as weight in kilograms divided by the square of the height in meters. The waist circumference (cm) was measured midway between the costal margin and the iliac crest at the end of a normal expiration. The blood pressure was measured on the right arm after a rest of ≥ 5 min using an automatic manometer with an appropriate cuff size. After overnight fasting, early morning blood samples were drawn from the antecubital vein into vacuum tubes and subsequently analyzed in the central, certified laboratory of the Asan Medical Center. Measurements included the concentrations of fasting plasma glucose, uric acid, creatinine, C-reactive protein (CRP), lipid parameters, and Hcy. Fasting total cholesterol, high

density lipoprotein cholesterol, low density lipoprotein cholesterol, triglyceride, uric acid, and creatinine levels were measured by the enzymatic colorimetric method using a Toshiba 200FR Neo chemistry autoanalyzer (Toshiba Medical System Co., Ltd., Tokyo, Japan). Fasting plasma glucose was measured by an enzymatic colorimetric method using a Toshiba 200 FR autoanalyzer (Toshiba). Ion-exchange high-performance liquid chromatography (Bio-Rad Laboratories, Inc., Hercules, CA, USA) was used to measure hemoglobin A1c levels. Serum CRP level was measured using a high-sensitivity latex particle-enhanced immunoturbidometric assay (Roche Diagnostics, Mannheim, Germany). Hcy concentrations were measured by a competitive immunoassay analyzed on the ADVIA Centaur (Bayer Diagnostics, Tarrytown, NY, USA) [15].

Obesity was defined as a body mass index $\geq$ 25 kg/m2 based on an Asian-specific cutoff point. Diabetes was defined as 1) a self-reported history of diabetes and/or treatment with dietary modification or use of anti-diabetic medication on the systemized questionnaire and 2) a fasting plasma glucose $\geq$ 126 mg/dL or a hemoglobin A1c level $\geq$ 6.5%. Hypertension was defined as blood pressure $\geq$ 140/90 mmHg, a self-reported history of hypertension, and/or use of anti-hypertensive medication. Hyperlipidemia was also defined as total cholesterol $\geq$ 240 mg/dL, a self-reported history of hyperlipidemia, and/or use of lipid-lowering treatment [15]. A family history of CAD was defined as having a first-degree relative of any age with CAD on the self-report questionnaire [18].

## CCTA image acquisition and analysis

CCTA was conducted using either a single-source (LightSpeed VCT, GE Medical Systems, Milwaukee, WI, USA) or dual-source 64-slice computed tomography scanner (Somatom Definition, Siemens Medical Solutions, Erlangen, Germany). Subjects with no contraindication to β-blockers and with an initial heart rate more than 65 beats/min are given bisoprolol 2.5 mg (Concor, Merck, Darmstadt, Germany) orally 1 hour before the CT examination. CT scanning was performed using the prospective ECG-triggering or the retrospective ECG-gating with ECG-based tube current modulation. Before contrast injection, two puffs (2.5 mg) of isosorbidedinitrate (Isoket spray, Schwarz Pharma, Monheim, Germany) were sprayed into the patient's oral cavity. During CCTA acquisition, 60–80 mL of iodinated contrast (Iomeron 400, Bracco, Milan, Italy) was injected at 4mL/s, followed by a 40 mL of saline flush. A region of interest of bolus-tracking technique was placed in the ascending aorta, and image acquisition was automatically triggered once a selected threshold (100 Hounsfield units) had been reached. A standard scanning protocol was used, and the tube voltage and tube current-time product were adjusted according to the patient's body size as follows: 100 kVp or 120 kVp tube voltage; 240 to 400 mAs per rotation (dual-source CT); and 400 to 800 mA (64-slice CT) tube current [15,19].

CCTA scans were analyzed using a dedicated workstation (Advantage Workstation, GE; or Volume Wizard, Siemens) by experienced cardiovascular radiologists (DHY, JWK, and THL) according to the guidelines of the Society of Cardiovascular Computed Tomography [20]. A coronary artery calcium score (CACS) was measured as previously described [21], and was categorized by scores of 0, 1 to 10, 11 to 100, 101 to 400, and > 400. Plaques were defined as structures > 1 mm$^2$ within and/or adjacent to the vessel lumen, which could be clearly distinguished from the lumen and surrounding pericardial tissue. Plaques containing calcified tissue involving > 50% of the plaque area (density > 130 Hounsfield units) were classified as calcified. Plaques with < 50% calcium in the plaque area were classified as mixed. Plaques without any calcium were classified as non-calcified [22]. The contrast-enhanced portion of the coronary lumen was semi-automatically traced at the site of maximal stenosis and compared with

the mean value of the proximal and distal reference sites. Significant stenosis was defined as a diameter stenosis ≥ 50% [15]. The overall plaque burden was determined from coronary artery plaque scores calculated based on segment involvement scores, segment stenosis scores, and modified Duke prognostic scores, as previously described [23].

## Statistical analysis

Categorical variables are expressed as frequencies with percentages, and continuous variables as means and standard deviations. Pearson's chi-squared test or Fisher's exact test was used for between-group comparisons of categorical variables, and one-way analysis of variance or Kruskal-Wallis test was used for between-group comparisons of numerical variables, as appropriate. A logistic regression model was used for univariable and multivariable analyses to evaluate the influence of Hcy level on the risk of subclinical coronary atherosclerosis on CCTA. We selected covariates in the multivariable model according to clinical importance as well as statistical significance, which included age, sex, body mass index, diabetes mellitus, hypertension, hyperlipidemia, creatinine, uric acid, current smoking, a family history of CAD, and high-sensitivity CRP ≥ 2 mg/L. Unadjusted and adjusted odds ratios (ORs) with 95% confidence intervals (CIs) for the logistic regression were calculated. All reported p values are two-sided, and $p < 0.05$ was considered statistically significant. Data manipulation and statistical analyses were performed using SPSS software (Version 18; SPSS Inc., Chicago, IL, USA).

## Results

### Population characteristics

The mean age of study participants was 53.8 ± 8.0 years and 2,202 (69.1%) participants were men. The baseline characteristics of the study participants according to the tertiles of Hcy are summarized in Table 1. The prevalence of men, diabetes mellitus, hypertension, current smoking, and obesity significantly increased with the Hcy category. Body mass index, waist conference, systolic and diastolic blood pressure, fasting blood glucose, total cholesterol, low-density lipoprotein cholesterol, triglycerides, triglyceride, creatinine, and uric acid levels were higher in increasing Hcy tertiles.

### CCTA findings

Table 2 shows the CCTA findings according to the Hcy tertiles. The mean CACS of study participants was 39.2 ± 142.7. There were significant differences in CACS according to the Hcy tertiles ($p < 0.001$). A total of 121 (0.2%) coronary segments were not interpretable due to artifacts. Any coronary atherosclerotic, calcified, non-calcified, and mixed plaques were detected in 1,313 (41.2%), 892 (28.0%), 555 (17.4%) and 242 (7.6%) individuals, respectively. The prevalence of any atherosclerotic, calcified, non-calcified, or mixed plaque increased with the Hcy tertiles ($p$ for all $< 0.05$). In addition, segment involvement score, segment stenosis score, and modified Duke prognostic score increased with the Hcy tertiles ($p$ for all $< 0.001$). Of the study participants, 230 (7.2%) had significant coronary arteries stenosis ($\geq 50\%$ diameter stenosis) in at least one coronary artery on CCTA. Significant stenosis in the left main, left anterior descending, left circumflex artery, and right coronary arteries was observed in 9 (0.3%), 150 (4.7%), 76 (2.4%), and 68 (2.1%) participants, respectively. Significant coronary arteries stenosis increased with the Hcy tertiles ($p = 0.006$).

**Table 1. Baseline characteristics of the study population according to the tertiles of homocysteine.**

| Characteristics | Overall (n = 3,186) | Homocysteine level | | | P value |
|---|---|---|---|---|---|
| | | Tertile 1 $\leq$ 10.4 µmol/L (n = 1,065) | Tertile 2 10.5–12.8 µmol/L (n = 1,061) | Tertile 3 $\geq$ 12.9 µmol/L (n = 1,060) | |
| Age, years | 53.8 ± 8.0 | 53.4 ± 7.6 | 54.2 ± 7.9 | 53.9 ± 8.4 | 0.054 |
| Male sex, no. (%) | 2,202 (69.1) | 455 (42.7) | 786 (74.1) | 961 (90.7) | < 0.001 |
| Body mass index, kg/m$^2$ | 24.6 ± 2.9 | 23.9 ± 2.8 | 24.7 ± 2.9 | 25.2 ± 2.9 | < 0.001 |
| Waist circumference, cm | 85.9 ± 8.4 | 82.9 ± 8.1 | 86.3 ± 8.2 | 88.5 ± 7.9 | < 0.001 |
| Systolic blood pressure, mmHg | 119.9 ± 13.2 | 118.1 ± 13.6 | 120.1 ± 12.8 | 121.6 ± 13.1 | < 0.001 |
| Diastolic blood pressure, mmHg | 76.0 ± 10.7 | 73.6 ± 10.4 | 76.3 ± 10.7 | 78.0 ± 10.6 | < 0.001 |
| Diabetes mellitus, no. (%) | 537 (16.9) | 158 (14.8) | 187 (17.6) | 192 (18.1) | 0.093 |
| Hypertension, no. (%) | 1,138 (35.7) | 289 (27.1) | 388 (36.6) | 461 (43.5) | < 0.001 |
| Hyperlipidemia, no. (%) | 1,036 (32.5) | 305 (28.6) | 366 (34.5) | 365 (34.4) | 0.004 |
| Current smoker, no. (%) | 729 (22.9) | 135 (12.7) | 259 (24.4) | 335 (31.6) | < 0.001 |
| Obesity, no. (%) | 1,375 (43.2) | 343 (32.2) | 481 (45.3) | 551 (52.0) | < 0.001 |
| Family history of coronary artery disease[a], no. (%) | 428 (13.4) | 137 (12.9) | 161 (15.2) | 130 (12.3) | 0.116 |
| Fasting blood glucose, mg/dL | 104.8 ± 20.5 | 102.8 ± 18.6 | 105.3 ± 21.7 | 106.2 ± 21.0 | < 0.001 |
| Total cholesterol, mg/dL | 196.2 ± 34.6 | 193.4 ± 34.5 | 196.6 ± 34.7 | 198.6 ± 34.6 | 0.003 |
| Low-density lipoprotein cholesterol, mg/dL | 122.4 ± 30.6 | 119.0 ± 30.0 | 123.0 ± 30.4 | 125.2 ± 31.1 | < 0.001 |
| High-density lipoprotein cholesterol, mg/dL | 53.4 ± 13.4 | 56.3 ± 13.9 | 52.9 ± 13.1 | 51.0 ± 12.5 | < 0.001 |
| Triglyceride, mg/dL | 131.2 ± 82.9 | 112.9 ± 66.5 | 134.5 ± 84.0 | 146.3 ± 92.7 | < 0.001 |
| Creatinine, mg/dL | 0.9 ± 0.2 | 0.8 ± 0.1 | 0.9 ± 0.2 | 1.0 ± 0.1 | < 0.001 |
| Uric acid, mg/dL | 5.6 ± 1.4 | 4.9 ± 1.3 | 5.7 ± 1.3 | 6.1 ± 1.4 | < 0.001 |
| High-sensitivity C-reactive protein $\geq$ 2 mg/L | 29 (0.9) | 10 (0.9) | 12 (1.1) | 7 (0.7) | 0.518 |

Values are shown as mean ± standard deviation or number (%).

[a]Coronary artery disease in a first-degree relative of any age

## Association between Hcy levels and subclinical coronary atherosclerosis

The association between Hcy level and subclinical atherosclerosis is shown in Table 3. Univariable analysis showed that the increasing tertiles of Hcy were significantly associated with coronary artery calcification (defined as CACS > 10), and any atherosclerotic, calcified, non-calcified, and mixed plaques. In addition, the third tertile of Hcy had a significant association with significant coronary artery stenosis compared to the first tertile of Hcy.

In multivariable analysis, after adjustment for cardiovascular risk factors (age, sex, body mass index, diabetes mellitus, hypertension, hyperlipidemia, current smoking, family history of CAD, creatinine level, uric acid level, and high-sensitivity CRP $\geq$ 2 mg/L), there were no

**Table 2. Comparison of coronary computed tomography angiographic findings according to the tertiles of homocysteine.**

| Variables | Overall | Homocysteine level | | | |
|---|---|---|---|---|---|
| | | Tertile 1 | Tertile 2 | Tertile 3 | P value |
| Mean coronary artery calcium score | 39.2 ± 142.7 | 23.0 ± 97.3 | 41.5 ± 138.9 | 53.0 ± 178.8 | < 0.001 |
| Coronary artery calcium score, no. (%) | | | | | < 0.001 |
| 0 | 2,083 (65.6) | 778 (73.2) | 691 (65.5) | 614 (58.0) | |
| 1–10 | 293 (9.2) | 82 (7.7) | 93 (8.8) | 118 (11.1) | |
| 11–100 | 511 (16.1) | 143 (13.5) | 160 (15.2) | 208 (19.6) | |
| 101–400 | 216 (6.8) | 51 (4.8) | 83 (7.9) | 82 (7.7) | |
| > 400 | 74 (2.3) | 9 (0.8) | 28 (2.7) | 37 (3.5) | |
| Any atherosclerotic plaque, no. (%) | 1,313 (41.2) | 357 (33.5) | 434 (40.9) | 522 (49.2) | < 0.001 |
| Plaque characteristics, no. (%) | | | | | |
| Calcified plaque | 892 (28.0) | 232 (21.8) | 300 (28.3) | 360 (34.0) | < 0.001 |
| Non-calcified plaque | 555 (17.4) | 164 (15.4) | 178 (16.8) | 213 (20.1) | 0.014 |
| Mixed plaque | 242 (7.6) | 50 (4.7) | 87 (8.2) | 105 (9.9) | < 0.001 |
| Segment involvement score | 1.0 ± 1.7 | 0.7 ± 1.4 | 1.0 ± 1.7 | 1.2 ± 1.8 | < 0.001 |
| Segment stenosis score | 0.5 ± 1.8 | 0.4 ± 1.4 | 0.5 ± 1.7 | 0.8 ± 2.2 | < 0.001 |
| Modified Duke prognostic score | 1.2 ± 0.6 | 1.1 ± 0.5 | 1.1 ± 0.5 | 1.2 ± 0.7 | < 0.001 |
| Significant stenosis, no. (%) | 230 (7.2) | 60 (5.6) | 73 (6.9) | 97 (9.2) | 0.006 |

Values are shown as mean ± standard deviation or number (%).

statistically significant differences in the adjusted ORs for coronary artery calcification (OR 1.07; 95% CI 0.83–1.37; p = 0.601), any atherosclerotic plaque (OR 1.06; 95% CI 0.85–1.32; p = 0.610), calcified plaques (OR 1.17; 95% CI 0.92–1.48; p = 0.199), non-calcified plaque (0.80; 95% CI 0.61–1.04; p = 0.089), and mixed plaques (OR 1.42; 95% CI 0.96–2.11; p = 0.077) in the third tertile of Hcy compared to the first tertile. In addition, the adjusted ORs for significant coronary artery stenosis (OR 0.92; 95% CI 0.63–1.36; p = 0.687) did not differ between the first and third tertiles.

## Discussion

The main finding of this study was that Hcy level was not associated with any subclinical coronary atherosclerosis on CCTA in asymptomatic individuals after adjusting for traditional cardiovascular risk factors.

It is still unclear whether Hcy is a cause or a marker of atherosclerotic vascular disease. The association between Hcy levels and subclinical atherosclerosis has been investigated in previous randomized studies using carotid intima-media thickness, aortic and coronary artery calcium, and pulse wave velocity. However, inconsistent results have been reported [24,25]. Moreover, it remains unknown whether Hcy levels are associated with subclinical coronary atherosclerosis. Since CCTA has been proved to provide comprehensive information regarding coronary atherosclerosis, including lesion location, plaque characteristics, and disease severity, we could identify and determine the relationship between cardiovascular risk factors and coronary atherosclerosis in previous studies [15,16,19]. Therefore, the present study aimed to evaluate the association between Hcy levels and the risk of subclinical coronary atherosclerosis through analysis from a large CCTA registry.

The present study showed that Hcy level was not a risk factor for any subclinical coronary atherosclerosis assessed by CCTA. Early observational studies and meta-analysis indicated that

**Table 3. Association between homocysteine levels and coronary computed tomography angiographic findings.**

| Variables | Univariable | | Multivariable | |
|---|---|---|---|---|
| | Odds ratio (95% CI) | P value | Odds ratio (95% CI) | P value |
| Coronary artery calcification[a] | | < 0.001 | | 0.566 |
| Tertile 1 (reference) | 1 | - | 1 | - |
| Tertile 2 | 1.46 (1.19–1.80) | < 0.001 | 0.95 (0.75–1.21) | 0.678 |
| Tertile 3 | 1.89 (1.55–2.31) | < 0.001 | 1.07 (0.83–1.37) | 0.601 |
| Any atherosclerotic plaque | | < 0.001 | | 0.062 |
| Tertile 1 (reference) | 1 | - | 1 | - |
| Tertile 2 | 1.37 (1.15–1.64) | < 0.001 | 0.85 (0.69–1.05) | 0.121 |
| Tertile 3 | 1.92 (1.62–2.29) | < 0.001 | 1.06 (0.85–1.32) | 0.610 |
| Calcified plaque | | < 0.001 | | 0.205 |
| Tertile 1 (reference) | 1 | - | 1 | - |
| Tertile 2 | 1.42 (1.16–1.73) | 0.001 | 0.98 (0.78–1.23) | 0.840 |
| Tertile 3 | 1.85 (1.52–2.24) | < 0.001 | 1.17 (0.92–1.48) | 0.199 |
| Non-calcified plaque | | 0.014 | | 0.066 |
| Tertile 1 (reference) | 1 | - | 1 | - |
| Tertile 2 | 1.11 (0.88–1.40) | 0.387 | 0.74 (0.58–0.96) | 0.022 |
| Tertile 3 | 1.38 (1.10–1.73) | 0.005 | 0.80 (0.61–1.04) | 0.089 |
| Mixed plaque | | < 0.001 | | 0.205 |
| Tertile 1 (reference) | 1 | - | 1 | - |
| Tertile 2 | 1.81 (1.27–2.60) | 0.001 | 1.30 (0.89–1.90) | 0.175 |
| Tertile 3 | 2.23 (1.58–3.16) | < 0.001 | 1.42 (0.96–2.11) | 0.077 |
| Significant stenosis | | 0.007 | | 0.578 |
| Tertile 1 (reference) | 1 | - | 1 | - |
| Tertile 2 | 1.24 (0.87–1.76) | 0.236 | 0.82 (0.56–1.20) | 0.313 |
| Tertile 3 | 1.69 (1.21–2.36) | 0.002 | 0.92 (0.63–1.36) | 0.687 |

CI = confidence interval

[a]Coronary artery calcification is defined as coronary artery calcium score > 10.

Covariates included in the multivariable model were age, sex, body mass index, diabetes mellitus, hypertension, hyperlipidemia, current smoking, family history of coronary artery disease, creatinine level, uric acid level, and high-sensitivity C-reactive protein ≥ 2 mg/L.

Hcy levels are associated with CVD [4–6]. However, the results of following trials showed conflicting results regarding the preventive effect of Hcy-lowering therapy with vitamin B supplementation on CVD [7–10]. Furthermore, previous randomized trials focused on the risk of CAD from Hcy failed to demonstrate the effect of Hcy-lowering for a role for primary and secondary prevention [8,11–13]. A recent review and meta-analysis of 15 randomized trials involving 71,422 patients with high Hcy levels regarding the prevention of CVD, found no differences between Hcy-lowering therapy and placebo groups regarding the incidence of myocardial infarction (risk ratio [RR] 1.02; 95% CI 0.95–1.10) or all-cause death (RR 1.01; 95% CI 0.86–1.06), except for a modest prevention of stroke, which was reduced by 10% (RR 0.90; 95% CI 0.82–0.99) [14]. In the current study, after adjustment for clinical and laboratory variables, no association was found between Hcy levels and subclinical coronary atherosclerosis on CCTA. Accordingly, in asymptomatic individuals with high Hcy levels, the risk of subclinical atherosclerosis was mostly explained by traditional cardiovascular risk factors. Therefore, based on previous and our findings, the reduction of traditional cardiovascular risk factors and lifestyle modification should be prioritized to reduce the risk of subclinical coronary atherosclerosis and prevent future coronary events in asymptomatic individuals.

The pathophysiological mechanisms of stroke are much more heterogeneous than those of CAD. Various causes, such as atherosclerosis, thromboembolic events, and bleeding may underlie cerebrovascular disease, which involve both large and small cerebral vessels. In contrast, CAD is mainly related to a local atherothrombotic process in relatively large coronary vessels [24]. In addition, previous studies observed that the neurologic system might be more vulnerable to the damaging effects of Hcy [26,27]. These different pathophysiologic mechanisms may account for the different risk of Hcy between the two conditions. However, further research is needed to elucidate the differences in the effects of Hcy between cerebrovascular disease and CAD.

Our study has several limitations. First, the present study was based in a single center. Moreover, because study participants voluntarily went to the hospital for general health examination, there was a potential for selection bias. Second, our study relied on self-reported past medical history. The reliability of self-reported data compared with objective sources could vary by study population and by diagnosis, which might affect obtained findings [28]. Third, our study did not include levels of B vitamins. However, vitamin B supplementation did not reduce the risk of CAD events [8,11–13]. Fourth, calcified plaques and higher CACS may lead to overestimation of significant coronary arteries stenosis. Fifth, the study population was exclusively Korean. Therefore, the generalization of obtained findings to other ethnic groups may be limited. Finally, the use of CCTA in asymptomatic individuals has not yet been justified despite advanced techniques for reducing the shortcomings of CCTA [29].

## Conclusions

In this large observational study with asymptomatic subjects undergoing CCTA, Hcy levels were not associated with an increased risk of subclinical coronary atherosclerosis. These findings should be further investigated and validated in additional studies.

## Author Contributions

**Conceptualization:** Sangwoo Park, Gyung-Min Park, Jinhee Ha, Young-Rak Cho.

**Data curation:** Gyung-Min Park, Dong Hyun Yang, Joon-Won Kang, Tae-Hwan Lim, Hong-Kyu Kim, Jaewon Choe.

**Formal analysis:** Sangwoo Park, Gyung-Min Park, Eun Ji Park.

**Funding acquisition:** Gyung-Min Park.

**Investigation:** Sangwoo Park, Jinhee Ha, Young-Rak Cho, Jae-Hyung Roh, Yujin Yang, Ki-Bum Won, Yong-Giun Kim.

**Methodology:** Gyung-Min Park.

**Supervision:** Seung-Whan Lee, Young-Hak Kim.

**Validation:** Gyung-Min Park.

**Writing – original draft:** Sangwoo Park, Gyung-Min Park, Jinhee Ha.

**Writing – review & editing:** Sangwoo Park, Gyung-Min Park, Jinhee Ha, Soe Hee Ann, Yong-Giun Kim, Shin-Jae Kim, Sang-Gon Lee.

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
