## [Decision Letter · Decision Letter 0]

18 Feb 2020

PONE-D-19-35017

Homocysteine is not a risk factor for subclinical coronary atherosclerosis in asymptomatic individuals

PLOS ONE

Dear Dr. Park,

Thank you for submitting your manuscript to PLOS ONE. After careful consideration, we feel that it has merit but does not fully meet PLOS ONE’s publication criteria as it currently stands. Therefore, we invite you to submit a revised version of the manuscript that addresses the points raised during the review process.

We would appreciate receiving your revised manuscript by Apr 03 2020 11:59PM. To enhance the reproducibility of your results, we recommend that if applicable you deposit your laboratory protocols in protocols.io, where a protocol can be assigned its own identifier (DOI) such that it can be cited independently in the future. For instructions see: http://journals.plos.org/plosone/s/submission-guidelines#loc-laboratory-protocols

We look forward to receiving your revised manuscript.

Kind regards,

Timir Paul

Academic Editor

PLOS ONE

Journal Requirements:

- Park, Gyung-Min, et al. "Prediabetes is not a risk factor for subclinical coronary atherosclerosis." International journal of cardiology 243 (2017): 479-484.

- Lim, Doo-Ho, et al. "Serum uric acid level and subclinical coronary atherosclerosis in asymptomatic individuals: An observational cohort study." Atherosclerosis 288 (2019): 112-117.

 The text that needs to be addressed involves some sentences of the Introduction and of the Discussion.

In your revision ensure you cite all your sources (including your own works), and quote or rephrase any duplicated text outside the methods section. Further consideration is dependent on these concerns being addressed.

3. In your Methods section, please provide additional information about the participant recruitment method and the demographic details of your participants. Please ensure you have provided sufficient details to replicate the analyses such as: a) the recruitment date range (month and year), b) a description of any inclusion/exclusion criteria that were applied to participant recruitment, c) a table of relevant demographic details, d) a statement as to whether your sample can be considered representative of a larger population, e) a description of how participants were recruited, and f) descriptions of where participants were recruited and where the research took place.

Reviewers' comments:

Reviewer's Responses to Questions

**Comments to the Author**

1. Is the manuscript technically sound, and do the data support the conclusions?

Reviewer #1: Yes

Reviewer #2: Yes

2. Has the statistical analysis been performed appropriately and rigorously? 

Reviewer #1: Yes

Reviewer #2: Yes

3. Have the authors made all data underlying the findings in their manuscript fully available?

Reviewer #1: Yes

Reviewer #2: Yes

4. Is the manuscript presented in an intelligible fashion and written in standard English?

Reviewer #1: Yes

Reviewer #2: Yes

5. Review Comments to the Author

Reviewer #1: This manuscript is well written. In this manuscript authors have described the effect of homocysteine on sublcinical atherosclerosis in asymptomactic Korean population. These patients underwent Coronary CT as part of their general exam and then they were evaluated for homocysteine levels as well. Authors did not find any significant relation between homocysteine and sub clinical atherosclerosis. Studies have not shown any relationship between homocysteine lower therapies and reduction in adverse cardiac outcomes. This study adds to existing clinical data. However as authors described these results are not generalizable to other population yet and further studies are needed to test it. I would say this article gives enough clinical information and is good to be published

Reviewer #2: Park S. et al has presented data supporting no relation between homocysteine levels and risk of subclinical coronary atherosclerosis in asymptomatic individuals. Overall manuscript is well written and authors have accepted major limitations of the study except few which I would like to point out. It would be great if authors could revise and include them in the manuscript

1) Artifacts caused by CCTA especially blooming artifact caused by Calcium, inability to clearly delineate coronary artery disease in distal segments of the major epicardial coronary arteries. Author should also include methodology of image aquisition in detail as it is the major source of coronary information obtained for this research.

2) Another major limitation of the study is inclusion of self reported history of hypertension, hyperlipidemia or diabetes and then using that population in multivariate analysis to adjust the statistical relation between homocysteine levels and subclinical coronary atherosclerosis. Assuming patients having condition based on their word and not strictly based on clinical parameters / lab results can significantly confound the study results.

3) Structural heart disease is one of the exclusion criteria which should be defined further in detail.

4) Another exclusion criteria is "prior history of open heart surgery". Its not clear if author means coronary artery bypass grafting (CABG) or any cardiac surgery. Should be clearly stated as one should not assume it mean CABG.

6. PLOS authors have the option to publish the peer review history of their article (what does this mean?). If published, this will include your full peer review and any attached files.

Reviewer #1: No

Reviewer #2: No

---

## [Author Response · Author response to Decision Letter 0]

10 Mar 2020

In reply to the Editor and Reviewers

We thank the Editor and Reviewers' for time, effort, and previous comments.

 First above all, we would like to express our gratitude for providing us another opportunity to address our study. We appreciate the Editor and Reviewers' contribution, time, and invaluable comments, which were truly meaningful to the revision of our manuscript. We gratefully accepted the Editor and Reviewers' advice and revised the manuscript. We are submitting the revised manuscript, on which we provide a detailed list of changes in response to all comments of the Editor and Reviewers in a point by point fashion. We hope that you will find our paper suitable for publication in your journal, and we look forward to hearing from you.

Journal Requirements:

Authors’ response:

We appreciate the Editor’s valuable comments.

With the response to the Editor’s comments, we revised our manuscript according to PLOS ONE's style requirements

- Park, Gyung-Min, et al. "Prediabetes is not a risk factor for subclinical coronary atherosclerosis." International journal of cardiology 243 (2017): 479-484.

- Lim, Doo-Ho, et al. "Serum uric acid level and subclinical coronary atherosclerosis in asymptomatic individuals: An observational cohort study." Atherosclerosis 288 (2019): 112-117.

 The text that needs to be addressed involves some sentences of the Introduction and of the Discussion. In your revision ensure you cite all your sources (including your own works), and quote or rephrase any duplicated text outside the methods section. Further consideration is dependent on these concerns being addressed.

Authors’ response:

We appreciate the Editor’s meaningful comments.

With the response to the Editor’s comments, we revised the manuscript appropriately. We addressed and cited our previous works in in the Introduction and Discussion.

3. In your Methods section, please provide additional information about the participant recruitment method and the demographic details of your participants. Please ensure you have provided sufficient details to replicate the analyses such as: a) the recruitment date range (month and year), b) a description of any inclusion/exclusion criteria that were applied to participant recruitment, c) a table of relevant demographic details, d) a statement as to whether your sample can be considered representative of a larger population, e) a description of how participants were recruited, and f) descriptions of where participants were recruited and where the research took place.

Authors’ response:

We appreciate the Editor’s invaluable comments.

With the response to the Editor’s comments, we revised and provided sufficient details in the Figure 1 and Method section, as much as we can. Demographic details of the study population can be found in overall column of Table 1.

Reviewers' comments:

Reviewer #1: This manuscript is well written. In this manuscript authors have described the effect of homocysteine on subclinical atherosclerosis in asymptomatic Korean population. These patients underwent Coronary CT as part of their general exam and then they were evaluated for homocysteine levels as well. Authors did not find any significant relation between homocysteine and sub clinical atherosclerosis. Studies have not shown any relationship between homocysteine lower therapies and reduction in adverse cardiac outcomes. This study adds to existing clinical data. However as authors described these results are not generalizable to other population yet and further studies are needed to test it. I would say this article gives enough clinical information and is good to be published

Authors’ response:

Thank you very much for your encouraging comments.

Reviewer #2: Park S. et al has presented data supporting no relation between homocysteine levels and risk of subclinical coronary atherosclerosis in asymptomatic individuals. Overall manuscript is well written and authors have accepted major limitations of the study except few which I would like to point out. It would be great if authors could revise and include them in the manuscript.

1) Artifacts caused by CCTA especially blooming artifact caused by Calcium, inability to clearly delineate coronary artery disease in distal segments of the major epicardial coronary arteries. Author should also include methodology of image aquisition in detail as it is the major source of coronary information obtained for this research.

Authors’ response:

We appreciate the Reviewer’s meaningful comments.

We fully agreed with the Reviewer’s valuable comments. We omitted the methodology of image acquisition because there was significant overlapping text with our previous studies. However, as the Reviewer mentioned, it is important information for this study, so we have described it in the CCTA image acquisition and analysis section. And, with the response to the Reviewer’s comments, we added the following sentences in the Results.

CCTA image acquisition and analysis, Methods, page 7

CCTA was conducted using either single-source 64-slice CT (LightSpeed VCT, GE, Milwaukee, WI, USA) or dual-source CT (Somatom Definition, Siemens, Erlangen, Germany). Subjects with no contraindication to β-adrenergic blocking agents and with an initial heart rate greater than 65 beats per minute received an oral dose of 2.5 mg bisoprolol (Concor, Merck, Darmstadt, Germany) 1 hour before the CT examination. CT scanning was performed in the prospective ECG-triggering mode or the retrospective ECG-gating mode with ECG-based tube current modulation. Two puffs (2.5 mg) of isosorbidedinitrate (Isoket spray, Schwarz Pharma, Monheim, Germany) were sprayed into the patient’s oral cavity before contrast injection. During CCTA acquisition, 60-80 mL of iodinated contrast (Iomeron 400, Bracco, Milan, Italy) was injected at 4 mL/second, followed by a 40 mL saline flush. A region of interest was placed in the ascending aorta, and image acquisition was automatically initiated once a selected threshold (100 HU) had been reached using bolus tracking. A standard scanning protocol was used, and the tube voltage and tube current-time product were adjusted according to the patient’s body size as follows: 100 kVp or 120 kVp tube voltage; 240 to 400 mAs per rotation (dual-source CT); and 400 to 800 mA (64-slice CT) tube current.

All CCTA scans were analyzed using a dedicated workstation (Advantage Workstation, GE; or Volume Wizard, Siemens) by experienced cardiovascular radiologists (DHY, JWK, and THL).

CCTA findings, Results, page 12

A total of 121 (0.2%) coronary segments were not interpretable due to artifacts.

2) Another major limitation of the study is inclusion of self reported history of hypertension, hyperlipidemia or diabetes and then using that population in multivariate analysis to adjust the statistical relation between homocysteine levels and subclinical coronary atherosclerosis. Assuming patients having condition based on their word and not strictly based on clinical parameters / lab results can significantly confound the study results.

Authors’ response:

We appreciate the Reviewer’s invaluable comments.

We fully agreed with the Reviewer’s comments. The reliability of self-reported data compared with objective sources could vary by study population and by diagnosis. We acknowledge the limitation of self-reported data and described it in the Limitation section.

Limitations, page 18

Second, our study relied on self-reported past medical history. The reliability of self-reported data compared with objective sources could vary by study population and by diagnosis, which might affect obtained findings.

3) Structural heart disease is one of the exclusion criteria which should be defined further in detail.

Authors’ response:

We appreciate the Reviewer’s valuable comments.

We presented the frequency of structural heart disease in detail, i.e., hypertrophic cardiomyopathy (n=33), moderate to severe valvular heart disease (n=11), atrial septal defect (n=2), dilated cardiomyopathy (n=1), myxoma (n=1), and dextrocardia (n =1). With the response to the Reviewer’s comments, revised sentences in the Methods.

Methods, page 5

We excluded subjects with 1) unmeasured Hcy; 2) a previous history of angina or myocardial infarction; 3) abnormal rest electrocardiographic results, i.e., pathological Q waves, ischemic ST segments or T wave changes, or left bundle-branch blocks; 4) incomplete medical records; 5) structural heart diseases, i.e., hypertrophic cardiomyopathy, moderate to severe valvular heart disease, atrial septal defect, dilated cardiomyopathy, myxoma, or dextrocardia; 6) a prior history of any open heart surgery or percutaneous coronary intervention; 7) a previous cardiac procedure; or 8) renal insufficiency (creatinine > 1.5 mg/dL). Finally, 3186 subjects were enrolled (Fig 1).

4) Another exclusion criteria is "prior history of open heart surgery". Its not clear if author means coronary artery bypass grafting (CABG) or any cardiac surgery. Should be clearly stated as one should not assume it mean CABG.

Authors’ response:

We appreciate the Reviewer’s meaningful comments.

We apologize to have caused the confusion. With the response to the Reviewer’s comments, revised sentences in the Methods.

Methods, page 5

We excluded subjects with 1) unmeasured Hcy; 2) a previous history of angina or myocardial infarction; 3) abnormal rest electrocardiographic results, i.e., pathological Q waves, ischemic ST segments or T wave changes, or left bundle-branch blocks; 4) incomplete medical records; 5) structural heart diseases, i.e., hypertrophic cardiomyopathy, moderate to severe valvular heart disease, atrial septal defect, dilated cardiomyopathy, myxoma, or dextrocardia; 6) a prior history of any open heart surgery or percutaneous coronary intervention; 7) a previous cardiac procedure; or 8) renal insufficiency (creatinine > 1.5 mg/dL). Finally, 3186 subjects were enrolled (Fig 1).

---

## [Editor Report · Decision Letter 1]

24 Mar 2020

Homocysteine is not a risk factor for subclinical coronary atherosclerosis in asymptomatic individuals

PONE-D-19-35017R1

Dear Dr. Park,

We are pleased to inform you that your manuscript has been judged scientifically suitable for publication and will be formally accepted for publication once it complies with all outstanding technical requirements.

With kind regards,

Timir Paul

Academic Editor

PLOS ONE

Additional Editor Comments (optional):

No further revision required

---

## [Editor Report · Acceptance letter]

26 Mar 2020

PONE-D-19-35017R1 

Homocysteine is not a risk factor for subclinical coronary atherosclerosis in asymptomatic individuals 

Dear Dr. Park:

I am pleased to inform you that your manuscript has been deemed suitable for publication in PLOS ONE. Congratulations! Your manuscript is now with our production department. 

With kind regards,

on behalf of

Dr. Timir Paul 

Academic Editor

PLOS ONE